# Isolation and Identification of Bovine Preadipocytes and Screening of MicroRNAs Associated with Adipogenesis

**DOI:** 10.3390/ani10050818

**Published:** 2020-05-09

**Authors:** Xiang Yu, Xibi Fang, Ming Gao, Jiaqi Mi, Xiuqi Zhang, Lixin Xia, Zhihui Zhao, Elke Albrecht, Steffen Maak, Runjun Yang

**Affiliations:** 1College of Animal Sciences, Jilin University, Changchun 130062, Jilin Province, China; fisheryx@163.com (X.Y.); fangxibi@jlu.edu.cn (X.F.); gaoming930410@126.com (M.G.); mijq18@mails.jlu.edu.cn (J.M.); xqzhang7@163.com (X.Z.); xialx18@mails.jlu.edu.cn (L.X.); 2College of Agriculture, Guangdong Ocean University, Zhanjiang 524088, Guangdong Province, China; zhzhao@jlu.edu.cn; 3Institute of Muscle Biology and Growth, Leibniz Institute for Farm Animal Biology (FBN), 18196 Dummerstorf, Germany; elke.albrecht@fbn-dummerstorf.de (E.A.); maak@fbn-dummerstorf.de (S.M.)

**Keywords:** bovine preadipocyte, differentiation, microRNA, lipid metabolism

## Abstract

**Simple Summary:**

Promoting fat deposition in beef cattle has been a focus of modern animal breeding research. However, previous researchers have not examined the mechanism of adipogenesis in much detail. MicroRNAs (miRNAs) are small noncoding RNAs that play a pivotal role in adipogenesis. In this study, to explore the molecular regulatory mechanism of adipocyte differentiation and formation, bovine preadipocytes were isolated and induced into adipocytes, and then the expression patterns of miRNAs between preadipocytes and adipocytes were detected through RNA sequencing. Deep sequence analysis has identified 78, 71, and 48 novel miRNAs and 497, 491, and 524 known miRNAs in the preadipocytes, and 44, 54, and 47 novel miRNAs and 519, 522, and 504 known miRNAs in the adipocytes. Among the annotated miRNAs, 131 bovine miRNAs were upregulated in adipocytes, and 119 bovine miRNAs were downregulated in adipocytes, such as bta-miR-3604, bta-miR-23b-3p, bta-miR-26a, and bta-miR-129-3p. Bovine target gene prediction results of these miRNAs show that numerous genes are associated with lipid metabolism. These results can provide both technical support and a research basis for promoting bovine adipocyte fat deposition.

**Abstract:**

The elucidation of the mechanisms of preadipocyte differentiation and fat accumulation in adipocytes is a major work in beef cattle breeding. As important post-transcriptional regulators, microRNAs (miRNAs) take part in cell proliferation, differentiation, apoptosis, and fat metabolism through binding seed sites of targeting mRNAs. The aim of this study was to isolate and identify bovine preadipocytes and screen miRNAs associated with adipogenesis. Bovine preadipocytes were isolated from subcutaneous fatty tissue and induced to differentiate into adipocytes. Verification of preadipocytes and adipocytes was performed by qRT-PCR (real-time quantitative reverse transcription PCR), Oil Red O staining, and immunofluorescence staining. Total RNA was extracted for small RNA sequencing. The sequencing data showed that 131 miRNAs were highly expressed in adipocytes, and 119 miRNAs were highly expressed in preadipocytes. Stem–loop qPCR (stem–loop quantitative real-time PCR) results showed that the expression patterns of 11 miRNAs were consistent with the sequencing results (miR-149-5p, miR-24-3p, miR-199a-5p, miR-33a, etc.). According to KEGG pathway and Gene Ontology (GO) analyses, multiple predicted target genes were associated with lipid metabolism. In summary, this study provides a protocol of isolating bovine preadipocytes and screening various differently expressed miRNAs during preadipocyte differentiation.

## 1. Introduction

In terms of livestock products, beef is popular for its high protein and low fat contents. Improving beef quality is one of the main focuses of beef cattle breeding. The sensory indicators for evaluating beef quality are mainly that of the marbling pattern, the tenderness, the juiciness, and the meat color [1]. Fat content is the most important factor affecting beef quality sensory indicators. Adipose tissue, which usually locates in subcutaneous fat, visceral fat, and muscle, is mainly composed of adipocytes [2,3]. The order of fat deposition in animal ontogenesis is generally visceral fat first, followed by subcutaneous fat (SCF), and finally intramuscular fat (IMF) [4,5]. SCF affects carcass surface fat coverage rate, and IMF affects marbling and tenderness. However, it is difficult to achieve high-purity IMF. The isolation of primary preadipocytes from SCF in vitro could contribute to further studies for bovine fat deposition and beef quality improvement. In this study, we discussed the method of isolating the primary preadipocytes, and optimized the induction conditions to improve the induction efficiency.

Adipogenesis is a process of differentiation from mesenchymal stem cells (MSCs) to preadipocytes, and then to adipocytes [6]. The differentiation of adipocytes is a biologically complicated pathway that is regulated by transcription factors, signaling pathways, hormones, and other regulators. Peroxisome proliferator-activated receptor-γ (PPARγ) regulates the transcription of target genes by connecting the reaction components of the promoter [7]. CCAAT/enhancer-binding protein-α (CEBP/α) is involved in transcriptional regulation of genes to promote differentiation [8]. The Wnt (wingless-type MMTV integration site) signaling pathway regulates the activity of adipogenesis through a classic approach (β-dependent) and a nonclassic approach (β-independent) [9,10]. The MAPK (mitogen-activated protein kinases) signaling pathway activates enzymes to regulate preadipocyte differentiation [11]. Thus, it is important to study the gene expression pattern, proliferation, and differentiation potential of adipocytes at a molecular level. In our study, the expression levels of genes associated with adipogenesis were detected in order to verify the differentiation of preadipocytes. By treating the cells with a cocktail medium, including dexamethasone (DEX), insulin, 3-isobutyl-1-methylxanthine (IBMX), and rosiglitazone, bovine preadipocytes can differentiate into mature adipocytes. However, different species have different sensitivities to the concentration of the induction reagent. This study briefly discussed the effects of different media and different concentrations of reagents on induction efficiency.

MicroRNAs (miRNAs) are small noncoding RNA molecules that can regulate target gene expression patterns through binding of the seed sites [12]. In the last few years, developments in RNA sequencing (RNA-Seq) technology have made it possible to efficiently identify the miRNAs that regulate fat deposition via a high-throughput approach. Medrano was the first to contrast transcriptomic profiles between bovine gland tissue and somatic cells using RNA-Seq technology [13]. Previous studies have screened and verified three miRNAs annotated to lipid metabolism in primary mammary epithelial cells. In addition, some transcriptome changes have been found during bovine preadipocyte differentiation in Qinchuan cattle [14]. RNA-Seq also has been used to study the bovine gene expression profile associated with reproduction [15,16]. Due to its high sensitivity in identifying differential gene expression profiles and in quantitative analysis of the transcriptome, RNA-Seq has been used in various livestock, including swine [17], sheep [18], and chickens [19].

The present study aimed to establish a standard for isolating and identifying bovine preadipocytes in vitro and for screening miRNAs associated with adipogenesis. Small RNA-Seq was used to analyze the different abundance of miRNAs between preadipocytes and adipocytes. The results can provide both technical support and a research basis for promoting bovine preadipocyte differentiation and fat deposition.

## 2. Materials and Methods

### 2.1. Ethics Statement

Animal experiments were performed in strict accordance with the Guide for the Care and Use of Laboratory Animals by the Animal Health and Use Committee of Jilin University (permit number: SYXK (Ji) pzpx20181227083).

### 2.2. Isolation, Culture, and Induction of Bovine Primary Preadipocytes

#### 2.2.1. Isolation and Culture of Bovine Primary Preadipocytes

Subcutaneous adipose tissue was aseptically detached from newborn Simmental calves and kept in a glass medium bottle containing phosphate-buffered saline (PBS; HyClone, Logan, UT, USA) and gentamicin/amphotericin B (Gibco, Shanghai, China). The sample was soaked in 75% ethyl alcohol (Beijing Chemical Works, Beijing, China) for 30 s and washed with PBS. Then, blood vessels and connective tissues were dissected from the adipose tissue with scissors and tweezers. Next, the tissue was minced into fine pieces and mixed with an equal volume of collagenase type I (Gibco, Shanghai, China) and neutral protease (Roche, Basel, Switzerland). The mixture was shaken in a water bath for 1 h at 37 °C, and the enzyme reaction was then ended by the addition of equal volumes of Dulbecco’s Modified Eagle’s Medium and Nutrient Mixture F-12 (DMEM/F12; Hyclone, Logan, UT, USA) supplemented with 10% fetal bovine serum (FBS; Hyclone, Logan, UT, USA). The digested product was filtered through a 200-μm mesh sieve and centrifuged at 300 g/min for 10 min. The supernatant was discarded, and the sediment was resuspended with culture medium at an optimum density. The cell suspension was seeded in a 10 cm^2^ dish (Thermo Scientific, Shanghai, China) and incubated in an atmosphere of 5% CO_2_ at 37 °C. After 24 h, the cells were washed with PBS to discard residual tissue and dead cells. When the density reached 80%, the cells were passaged using trypsin (Gibco, Shanghai, China). For explant method, the operations before adding digestive enzymes were the same as the enzyme digestion. And the tissue blocks were placed in the culture dish with a few medium.

#### 2.2.2. Cell Proliferation Assay

Cells in the logarithmic phase were collected, adjusted to the optimum concentration, and then seeded in 96-well plates (Falcon, Franklin Lake, NJ, USA). To each well, 100 μL medium and 10 μL Cell Titer 96^®^ AQ One Solution (Promega, Madison, Wisconsin, USA) were added, and the cells were incubated at 37 °C for 4 h. Cell proliferation was measured with a microplate reader (Bio Tek Instruments EON, Madison, Vermont, USA) at 490 nm every two days.

#### 2.2.3. Adipogenic Differentiation of Preadipocytes

Preadipocytes were seeded into six-well plates (Falcon, Franklin Lake, NJ, USA) at 60% confluence in growth medium (DMEM/F12, 10% FBS, 1% antibodies). When the cells reached the contact inhibition stage, they were cultured in induction medium (growth medium, 10 μg/mL insulin (Sigma, Shanghai, China), 0.5 mM 1-methyl-3-isobutylxanthine (Sigma, Shanghai, China), 1.0 μM dexamethasone (Sigma, Shanghai, China), and 0.5 μM Rosiglitazone (Sigma, Shanghai, China). Two days later, the medium was replaced with maintenance medium (growth medium with 10 μg/mL insulin), with continued medium replacement every three days. Control cells were cultured without an inducer. Three different media were used to culture preadipocytes: 10% FBS in DMEM/F12; 2% serum replacement (SR; Sigma, Shanghai, China) and 0.5% CTS™ StemPro™ MSC (SFM; Gibco, Shanghai, China) in DMEM/F12; and COSMEDIUM H001 (COSMO, Tokyo, Japan).

### 2.3. Oil Red O Staining

Oil Red O staining is currently the most common method for estimating lipid accumulation. Briefly, the cells were gently rinsed twice with cold PBS and then fixed in paraformaldehyde (Biosharp, Hefei, China) at 37 °C for 30 min. Next, the cells were washed with PBS and stained with Oil Red O (Sigma, Shanghai, China) at room temperature for 20 min. After staining, the cells were soaked in 60% isopropanol (Beijing Chemical Words, Beijing, China) for 30 s and dyed in the dark for 5 min with hematoxylin. Then, the cells were observed under a microscope (Olympus, Tokyo, Japan) within an hour.

### 2.4. QRT-PCR Analysis of Adipogenic Marker Genes

Total RNA was extracted from the cells at different stages (induction for 3 days, 6 days, and 9 days, and control) using TRIzol reagent (Invitrogen, California, USA), and the concentration was measured using a spectrophotometer (Bio Tek Instruments EON, Vermont, USA). A total of 1 μg RNA was reversely transcribed into cDNA using a Prime Script^TM^ RT reagent kit (Takara, Dalian, China). The expression levels of adipocyte-specific genes, including *LPL*, *PPARγ*, *C/EBPα*, *FABP4*, *FASN*, and *DLK1*, were measured via quantitative reverse transcription PCR (qRT-PCR) with FastStart Universal SYBR^®^ Green Master (ROX) (Roche, Basel, Switzerland). The primers are shown in Table 1. The following PCR amplification conditions were used: 95 °C for 5 min; 40 cycles of 95 °C for 30 s, 60 °C for 30 s, and 72 °C for 20 s. β-Actin was selected as the housekeeping gene to normalize the quotient.

### 2.5. Immunofluorescence Staining

Cells were fixed with 4% paraformaldehyde for 30 min and permeabilized in 0.1% Triton X-100 (BIO BASIN, Shanghai, China) for approximately 20 min. Then, the cells were blocked with 1% bovine serum albumin (BSA; Biosciences, Franklin Lakes, New Jersey, USA) for 30 min and stained with Anti-DLK1 antibody (Bioss, bs-1973R, Beijing, China), Anti-PPAR Gamma antibody (Bioss, bs-4590R, Beijing, China), and Anti-lipoprotein lipase antibody (Bioss, bs-1973R, Beijing, China) at 37 °C for 1 h, followed by incubation with secondary antibody (Bioss, bs-0294R, Beijing, China). The cell nuclei were stained with DAPI (Bioss, Beijing, China) for 5 min. All images were captured with a fluorescence microscope (Nikon, Tokyo, Japan).

### 2.6. Analysis of Triglyceride Content

The total triglyceride content was detected with a triglyceride kit (KeyGEN BioTECH, Beijing, China) according to the manufacturer’s protocol. Briefly, the cells were washed with PBS twice and lysed for 10 min at room temperature. Then, the lysate was boiled at 70 °C for 10 min and centrifuged at 2000 rpm. Detection reagent R1 and detection reagent R2 were mixed at a 4:1 ratio. The absorbance was determined at 550 nm with a microplate reader. The cellular triglyceride content was calculated according to a standard curve.

### 2.7. Small RNA-seq and Analysis

Total RNA was extracted from adipocytes and preadipocytes using TRIzol reagent according to the manufacturer’s protocol. The RNA concentration and integrity were evaluated with agarose gel and a spectrophotometer (Nanodrop 2000, Thermo Scientific, Waltham, MA, USA). A total of 1 μg of RNA per sample was submitted to the Beijing Genomics Institute of Biotechnology (Wuhan) Co., Ltd. The sequence raw reads were filtered by removing adaptor contaminant, including 5’ primer contaminants, no-insert tags, oversized insertion tags, low-quality tags, poly A tags, and small tags. Then, AASRA [20] was used to map clean reads to the database of National Center for Biotechnology Information (NCBI) (version: GCF_000003055.6_Bos_taurus_UMD_3.1.1). All annotated sRNAs were classified according to the following order: MiRbase > pirnabank > snoRNA (human/plant) > Rfam > other sRNA. The known miRNAs were identified using miRBase18.0, and novel miRNA prediction was performed with miRDeep2 according to the characteristic hairpin structure of the miRNA precursor [21]. Transcripts Per Kilobase Million (TPM) was used to calculate the small RNA expression level to eliminate the influence of sequence discrepancy. The formula for determining TPM is shown as follows:$$\mathrm{TPM}=\frac{C*10^{6}}{N}$$
where “*C*” represents the miRNA count number in a sample, and *“N”* is the total read number mapped to the genome. For screening significantly differentially expressed SRNAs (DESs), a novel, previously proposed method based on the MA-plot was applied [22]. DEGs were defined as genes with more than two-fold expression differences and a Q-value ≤ 0.001. Hierarchical clustering for differentially expressed miRNAs was performed using a function heatmap in R.

### 2.8. Function Analysis of the Prediction Target Genes of Differentially Expressed miRNAs

To obtain more accurate results, RNAhybrid [23], miRanda, and TargetScan [24] were used to predict target genes. The results were filtered by free energy and score values. To find the DEGs that correspond to specific biological functions, Gene Ontology (GO) enrichment analysis and pathway enrichment analysis were performed with ‘GO: TermFinder’ (http://www.yeastgenome.org/help/analyze/go-term-finder) and KEGG [25] (the major public pathway-related database), respectively. For GO enrichment, the analysis mapped the target DEGs of differentially expressed miRNAs with the terms of the Gene Ontology database (http://www.geneontology.org/) and calculated the number of genes for each term. The *p*-value was corrected using the Bonferroni method [26], and a corrected *p*-value ≤ 0.05 was used as the threshold. The network between miRNAs and target genes was constructed with Cytoscape 3.7.1 (National Resource for Network Biology, Bethesda, MD, USA).

### 2.9. Validation of Differentially Expressed miRNAs

Eleven microRNAs, namely, bta-miR-193a-3p, bta-miR-199a-3p, bta-miR-199a-5p, bta-miR-199b, bta-miR-33a, bta-miR-29b, bta-miR-379, bta-miR-148a, bta-miR-449a, bta-miR-149-5p, bta-miR-24-3p, and bta-miR-23b-3p, were selected randomly from differentially expressed miRNAs between preadipocytes and adipocytes. RNAs were converted with miRNA-specific reverse transcription primers (Table 2). qRT-PCR quantification was performed following the manufacturer’s instructions with miRNA-specific primers and a universal primer. U6 was used as a control.

### 2.10. Statistical Analysis

Statistical analysis was performed with SPSS software (version: 22.0). The data are shown as the mean ± SD. Student’s *t*-test, one-way ANOVA, and multiple *t*-test analysis were used for comparisons. *p*-value < 0.05 was considered to indicate statistical significance. 

## 3. Results

### 3.1. The Isolation and Induction of Bovine Preadipocytes

#### 3.1.1. The Morphology and Proliferation of Primary Preadipocytes

Almost all of the cells digested by enzymes adhered to the Petri dish and were spindle-shaped after 6 h (Figure 1Ⅰa,Ⅰb). Observation of the tissue culture revealed that cell adherence and growth were slow. The tissue cells adhered to the wall of the flask and displayed a monolayer and a mosaic arrangement (Figure 1Ⅰc,Ⅰd). Cell proliferation assays showed that the bovine preadipocytes passed into the logarithmic phase until day 4, and the growth gate reached a peak at day 4.

#### 3.1.2. The Induction Efficiency of Preadipocytes under Different Conditions

Three induction media were used to induce adipogenic differentiation of bovine preadipocytes. Cells cultured with 10% FBS in DMEM/F12 proliferated rapidly and formed a dense layer (Figure 1Ⅱ). After 10 days of induction, small lipid drops formed, but bigger drops did not accumulate. Cells cultured with the COSMEDIUM H001 medium showed a fibroblast shape. There were no apparent lipid drops in the cytoplasm of induced cells (Figure 1Ⅱ). At later stage, obvious lipid droplets could be seen in the cells and converged into large lipid droplets over time when the preadipocytes proliferated in DMEM/F12 with 2% SR and 0.5% SFM. The adipocytes induced with 2% SR and 0.5% SFM had the highest induction efficiency, as determined by observing morphology and lipid droplet formation (Figure 1Ⅱ). Thus, all subsequent induction experiments were performed with 2% SR and 0.5% SFM in DMEM/F12.

#### 3.1.3. Cell Morphology and the Formation of Lipid Drops

Preadipocytes grew on the petri dish and developed fusiform or triangular tentacles. As the cells became more confluent (70%–80%), their morphology resembled that of fibroblasts. Most nuclei were located in the center of the cells. After seven days in maintenance differentiation medium, an oval or round shape was observed in the cytoplasm (Figure 2aI,aII,aIII), which varied in size, and bright spots were observed at the periphery of the nucleus. With prolonged differentiation time, the microdroplets gradually merged into large and bright drops, while the nucleus was squeezed to the side of the membrane (Figure 2aIV). The picture showed that the drops in differentiated cells were stained red in the induced group (induction for nine days), while the uninduced differentiation group showed no significant red lipid droplet. Lipid droplets stained with oil red O were extracted with isopropyl alcohol, and the OD value at 510 nm was determined (Figure 2aⅤ,aⅥ). The results show that the intracellular fat content in the induced differentiation group was significantly higher than that in the control group, which corresponded with the morphological observation.

#### 3.1.4. The Expression Levels of Adipogenesis Marker Genes During Preadipocytes Differentiation

According to the qRT-PCR results, the expression levels of the key genes associated with adipogenesis, including *PPARγ*, *C/EBPα*, *FABP4*, *FASN*, and *LPL*, were increased. The expression level of *PPARγ* peaked at day 9, while that of *C/EBPα* and *FABP4* peaked on day 6. In the late differentiation stage, the expression level of *PPARγ*, *FABP4*, and *LPL* significantly decreased (*p* < 0.05). As a marker gene of preadipocytes, *DLK1* was highly expressed in preadipocytes and scarcely expressed in adipocytes (Figure 2b).

#### 3.1.5. Immunofluorescence Staining and Detection of Cellar Triglyceride Content

Immunofluorescence staining was performed on preadipocytes and differentiated adipocytes. The results show that DLK1 is a protein specifically expressed in preadipocytes, and the fluorescence analysis exhibited higher expression levels in preadipocytes than in adipocytes. As a transcription factor, the fluorescence of PPARγ only expressed in the nucleus, whereas the fluorescence of LPL was expressed in the whole cell. Nevertheless, LPL and PPARγ showed no significant expression difference between preadipocytes and adipocytes (Figure 2c). Furthermore, the triglyceride synthesis in preadipocytes and differentiated adipocytes was determined. After seven days of induction, there was a 3-fold increase in the triglyceride content of preadipocytes compared with that in adipocytes. The results indicate that adipocyte differentiation and fat deposition increased significantly with the increase of induction time.

### 3.2. Analysis of miRNA Sequencing in Adipocytes at Different Stages of Differentiation

To investigate the involvement of miRNAs in the process of bovine preadipocyte differentiation, six miRNA sequencing libraries were generated using preadipocytes (b-PAD-1, b-PAD-2, and b-PAD-3) and differentiated adipocytes (b-AD-1, b-AD-2, and b-AD-3). Libraries were sequenced and constructed using the BGISEQ-500 small RNA deep sequencing technology and 18–30-base-long sequence reads were generated. Accordingly, 31.8 and 30.7 million mean reads of the biological triplicates were obtained from preadipocytes and adipocytes, respectively. After filtering out low-quality reads and empty adaptors, 27.9, 23.6, and 30.6 million clean reads were obtained from the preadipocyte libraries; meanwhile, 24.2, 24.6, and 28.4 million clean reads were obtained from the adipocyte libraries (shown in Table 3). Quality clean reads were mapped to the bovine reference genome for both the detection of known annotated miRNAs and the prediction of novel miRNAs. The analysis of sRNA length showed that most reads were 23 nt in the preadipocyte samples and similar in the adipocyte samples. The filtered clean tags were annotated using known small RNA databases, including miRBase, Rfam, and other small RNA, and all sample annotation percentages were more than 94%. A correlation heatmap showed that preadipocyte samples and adipocyte samples were correlated (Figure 3a). At the same time, hierarchical clustering was performed for differentially expressed miRNAs (Figure 3b). The results indicate that the samples had good uniformity among the three replicates. 

### 3.3. Annotation of Known miRNAs and Prediction of Novel miRNAs in Bovine Preadipocytes and Adipocytes

After comparison with databases of known small RNAs, approximately 500 known miRNAs were identified in each sample. The remaining unknown reads were predicted as novel miRNAs using the stem–loop structure of the miRNA precursor. According to the sets, 78, 71, and 48 novel miRNAs and 497, 491, and 524 known miRNAs were identified in the preadipocyte samples, and 44, 54, and 47 novel miRNAs and 519, 522, and 504 known miRNAs were identified in the adipocyte samples. Screening of differentially expressed sRNAs (DESs) is designed to find differentially expressed small RNAs among samples and to perform further analysis. A total of 250 miRNAs were differentially expressed between the two groups (Q-value ≤ 0.001). Among them, 119 miRNAs were highly expressed in bovine preadipocytes, and 131 miRNAs were highly expressed in bovine adipocytes (Figure 3c). A volcano plot showing the distribution of the differentially expressed miRNAs is presented in Figure 3d.

### 3.4. Functional Analysis of Target Genes and Stem–Loop qRT-PCR of miRNAs Related to Preadipocyte Differentiation

A total of 13,480 target genes of differently expression miRNAs were predicted using the RNAhybrid and miRanda databases. According to the results, many target genes are important adipogenesis genes, such as *FASN*, *C/EBPα*, *PPARγ*, *ADIPOQ*, *LPL*, *SERBP1*, *DLK1*, etc. *FASN* was predicted to be sponged by miR-24-3p, miR-197, miR-431, miR-409b, miR-11982, miR-10179-5p, miR-326, miR-6715, and miR-370. *PPARγ* was predicted to be sponged by miR-10179-5p. *C/EBPα* was predicted to be sponged by miR-10179-5p, miR-11976, miR-370, miR-1343-5p, and miR-2887. What is more, we found that miR-129-5p and miR-370, whose target gene was verified as *DLK1*, were also significantly expressed between preadipocytes and adipocytes. All of these results could indicate that differentially expressed RNAs are associated with bovine preadipocytes adipogenesis.

The top three biological process terms were mainly enriched in cellular processes, signaling, and metabolic processes. Considering the KEGG pathway analysis, a total of 44 pathways were enriched. Among these, 4.76% of the pathways were enriched in lipid metabolism, including the PPARγ signaling pathway, biosynthesis of unsaturated fatty acids, and fatty acid metabolism (Figure 3e). A total of 65 GO terms were annotated with the target genes, and 26 terms corresponded to biological processes, 19 to cellular components, and 20 to molecular functions (Figure 3f). A total of 4723 out of the 13,480 target genes were enriched in cell differentiation (*p* = 1.59 × 10^−8^), and 1140 out of the 13,480 target genes were enriched in cellular lipid metabolic processes. Both KEGG analysis and GO enrichment analysis showed that the differentially expressed miRNAs are indeed associated with adipogenesis.

Identical expression patterns of 11 miRNAs were observed between the stem–loop qRT-PCR and the RNA-seq results (Figure 4). The expression levels of miR-149-5p, miR-24-3p, and miR-23b-3p were higher in preadipocytes, and those of miR-193a-3p, miR-199a-3p, miR-199a-5p, miR-33a, miR-199b, miR-379, miR-148a, and miR-449a were higher in adipocytes. The expression patterns of the miRNAs were perfectly matched with the sequencing results. The network of miRNAs and some target genes are presented in Figure 4b.

The target genes of miR-149-5p that upregulated in preadipocytes were *ABHD4*, *CMTM3*, *PDGFRB*, *IGFBP5*, etc., as well as miR-23b-3p and miR-24b-3p. The target genes of miR-148 that downregulated in preadipocytes were *ABCA1*, *PCYT23*, *MAFB*, *SERPINE1*, *PRRG1*, etc., as well as miR-199b, miR-148a, miR-33a, miR-379, miR-29b, and miR-199a-3p, which promote adipocyte differentiation (Figure 5).

## 4. Discussion

The differentiation of preadipocytes into adipocytes is a process highly regulated by genes and signal transduction pathways. *Pref-1 (DLK1)* can accelerate preadipocyte proliferation and inhibit preadipocytes through two isoforms (Dlk1S and Dlk1M) [27]. Thus, *DLK1* has been found to be a landmark protein in preadipocytes, which is consistent with the results of our study. Furthermore, we found that miR-129-5p, whose target gene was verified as *DLK1*, was also significantly expressed between preadipocytes and adipocytes in the sequencing results. *PPARγ* is a ligand-activated transcription factor that regulates the expression of genes related to adipocyte proliferation and differentiation, lipid biosynthesis, and glucose metabolism [28]. *CEBP/α* belongs to the leucine zipper transcription factor family and is a transcriptional regulator of adipocyte differentiation; it also plays a key role in fat aggregation [29]. In our study, the expression levels of *PPARγ* and *CEBP/α* reached a peak during terminal differentiation. This illustrates that the two transcriptional regulatory factors have a pivotal role in adipogenesis.

The type and concentration of reagent are important for cell differentiation. In this study, adipose tissue was collected from perirenal adipose and subcutaneous fat tissue obtained from newborn cattle. A hormone cocktail that included insulin, IBMX, rosiglitazone/troglitazone, and dexamethasone was used to promote differentiation in vitro. The four reagents played different roles in the induction. Insulin not only promotes accumulation in adipocytes, but also has strong antiapoptotic activity by activating IGF-1 (an insulin-like growth factor receptor). Dexamethasone is an agonist for glucocorticoid synthesis as it stimulates the glucocorticoid receptor pathways [30]. IBMX is an inhibitor of cAMP phosphodiesterase by stimulating the c-amp-dependent protein kinase pathways. Rosiglitazone/troglitazone is an activator of the PPARγ receptor. Our study proved that the best induction medium is DMEM/F12 with 2% SR and 0.5% SFM, and induction reagents are 10 μg/mL insulin, 0.5 mM IBMX, 1.0 μM dexamethasone, and 0.5 μM Rosiglitazone.

With the development of sequence technology, miRNAs have been used for molecular breeding of beef cattle. An increasing number of studies have proposed that miRNAs participate in cell proliferation, differentiation, and apoptosis [31]. Some studies have found a potential role of miRNAs in lipid metabolism. Previous studies have also found that miR-103 can promote lipogenesis via sponging of the marker genes of adipogenesis [32]. In our study, the expression of miR-103 was also upregulated significantly in adipocytes. Some of the miRNAs filtered in our results have also been verified to be associated with lipid metabolism. Among these miRNAs, the sequence of miR-199a-5p is highly conserved in animals. MiR-199a-5p was higher expressed in the adipocytes in our study. Some evidence suggests that this miRNA could influence fat metabolism. Simultaneously, Kajimoto has shown a higher miR-199a-5p expression level in undifferentiated 3T3-L1 preadipocytes [33]. Furthermore, miR-148a was higher expressed in the adipocytes in our study. Previous research has indicated that miR-148a influences adipogenesis by regulating the target gene Wnt1 to inhibit the Wnt signaling pathway [34], and it has also been shown that it can inhibit the expression of LDLR to increase blood cholesterol levels [35]. We hypothesize that miR-199a-5p and miR-148a have the same effect on cattle, and subsequent verification is ongoing. All of the above indicates that the expression level of miRNAs could play an important part in fat metabolism. By comparing the different expression pattern of preadipocytes and adipocytes, our study screened more miRNAs that can regulate adipogenesis.

The biological interpretation of the target genes of differentially expressed miRNAs was investigated via KEGG pathway analysis. Some pathways related to fat regulation were significantly enriched, including the PPARγ signaling pathway, biosynthesis of unsaturated fatty acids, fatty acid metabolism, and focal adhesions. Focal adhesions are polyprotein structures that connect the intracellular cytoskeleton and extracellular matrix [36]. The downstream pathways of focal adhesions are also associated with fat metabolism. PAK1 (p21-activated kinase 1) is a key enzyme in focal adhesions and is connected to the downstream MAPK signaling pathway [37]. Considering that the MAPK signaling pathway is associated with energy metabolism, focal adhesions have a certain influence on lipid metabolism. In this study, a combined analysis of the predicted target genes of differentially expressed miRNAs and differentially expressed gene transcriptome data was also performed [14]. GO enrichment analysis showed that many genes are correlated with lipid metabolism. Three genes, namely, *ANGPTL8* (angiopoietin-like protein 8, cholesterol metabolism, ko04979), *NR1H3* (oxysterols receptor LXR-alpha, PPAR signaling pathway, ko03320), and *STAR* (steroidogenic acute regulatory protein, mitochondrial precursor, cholesterol metabolism, ko04979), were associated with fat deposition. The results show that 81.5% of the genes were annotated, which indicates that the differentially expressed miRNAs identified as regulators by the present study play important roles in the process of adipogenesis.

## 5. Conclusions

In our study, we characterized the miRNAs associated with adipogenesis in bovine preadipocytes. The sequencing results showed that 119 miRNAs (59 known miRNAs and novel 60 miRNAs) were downregulated and 131 miRNAs (114 known miRNAs and 17 novel miRNA) were upregulated. The expression profiles of 11 miRNAs verified by qRT-PCR, including miR-149-5p, miR-24-3p, miR-23b-3p miR-193a-3p, miR-199a-3p, miR-199a-5p, miR-33a, miR-199b, miR-379, miR-148a, and miR-449a, were the same to the sequencing data. Based on the KEGG pathway analysis, the predicted target genes were mainly enriched in signal transduction, transport and catabolism, and lipid metabolism. This work contributes to existing knowledge by providing an understanding of bovine adipogenesis at the molecular level.

## Figures and Tables

**Figure 1 animals-10-00818-f001:**
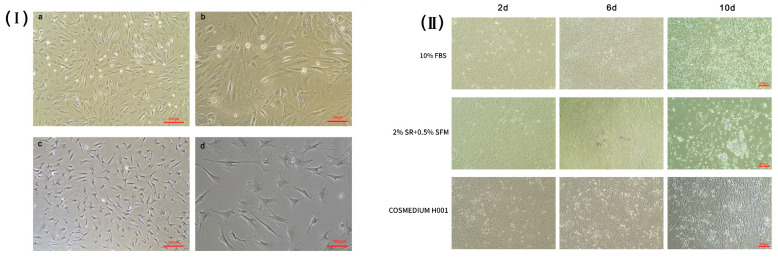
(**Ⅰ**) The morphology and proliferation of primary adipocytes isolated from tissue via the explant method (a,b) and the enzyme digestion method (c,d); (**Ⅱ**) The induction efficiency of preadipocytes under different conditions at different days. Induction conditions: A: 10% fetal bovine serum (FBS) in Dulbecco’s Modified Eagle’s Medium and Nutrient Mixture F-12 (DMEM/F12); B: 2% serum replacement (SR) + 0.5% SFM; C: COSMEDIUM H001.

**Figure 2 animals-10-00818-f002:**
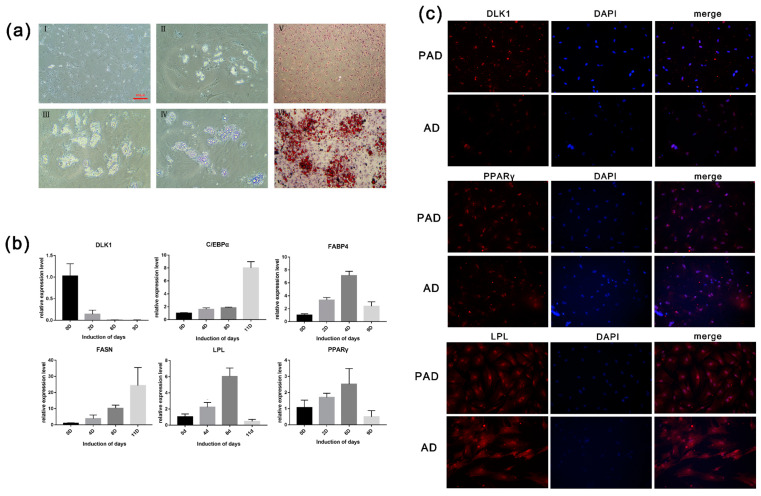
(**a**) Oil Red O staining of preadipocytes and adipocytes; (**b**) The mRNA expression levels of the marker genes of preadipocytes and adipocytes; (**c**) Immunofluorescence staining of preadipocytes and adipocytes. (AI: induction of day 2; AII: induction of day 6; AIII: induction of day 8; AIV: induction of day 11; AⅤ: Oil Red O staining of preadipocytes; AⅥ: Oil Red O staining of adipocytes). PAD, preadipocytes; AD, adipocytes.

**Figure 3 animals-10-00818-f003:**
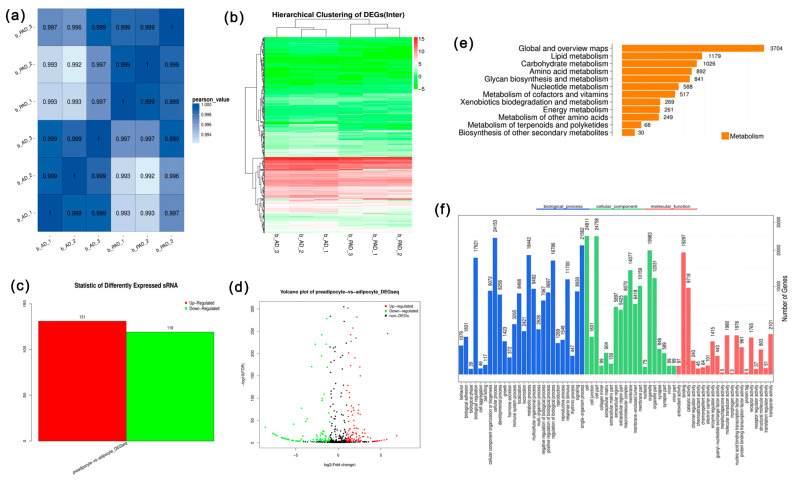
(**a**) Correlation analysis of all samples; (**b**) Hierarchical clustering of all samples; (**c**) Differentially expressed miRNAs between preadipocytes and adipocytes; (**d**) Volcano plot showing differential gene expression profiles of preadipocytes and adipocytes; (**e**) Lipid metabolism of the Kyoto Encyclopedia of Genes and Genomes (KEGG) pathway analysis of target genes; (**f**) Gene Ontology (GO) functional classification of target genes.

**Figure 4 animals-10-00818-f004:**
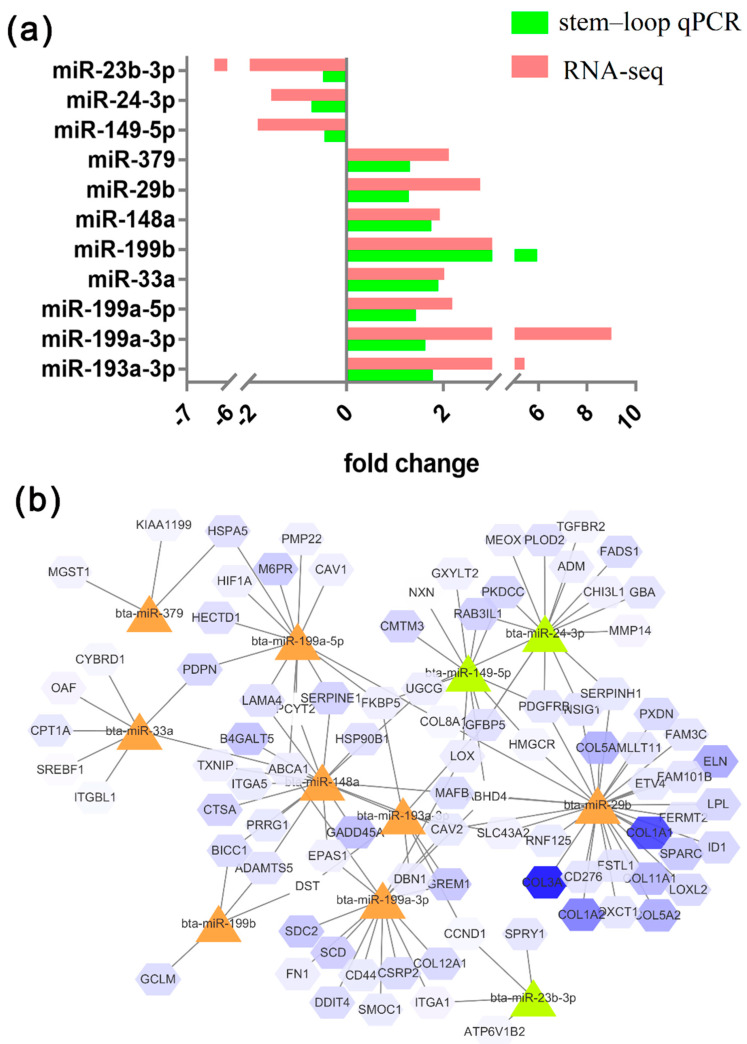
(**a**) Verification of sequencing results via stem–loop qPCR; (**b**) Network of differentially expressed miRNAs and their target genes.

**Figure 5 animals-10-00818-f005:**
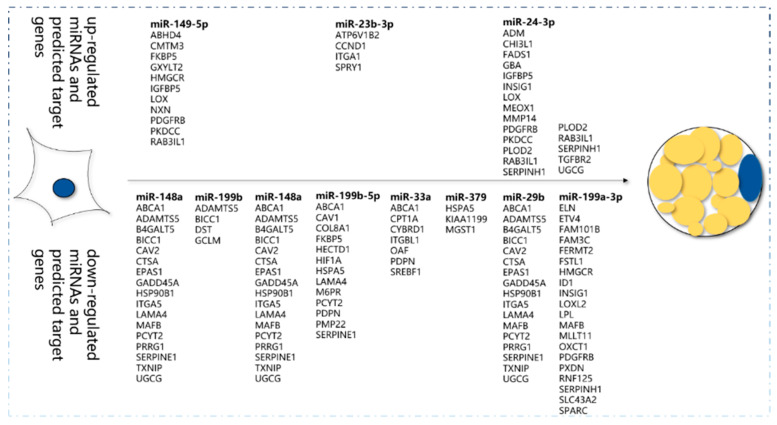
Candidate miRNAs and relevant target genes in bovine preadipocytes and differentiated adipocytes.

**Table 1 animals-10-00818-t001:** Primer sequences used for qRT-PCR.

Primer Name	Sequence (5′- 3′)	Amplicon Length (bp)
ACTB	for ATGCTTCTAGGCGGACTGTTA rev TGCCAATCTCATCTCGTTTTC	154
PPARγ	for TTCCACTCCGCACTATGA rev GATACAGGCTCCACTTTGA	117
DLK1	for CTTGCTCCTGCTGGCTTTCG rev ACCAGTGACCCTCTGTTTGG	146
FABP4	for GGATGGAAAATCAACCACCA rev TGGACAACGTATCCAGCAGA	174
C/EBPα	for TGGACAAGAACAGCAACGAGT rev GGTCATTGTCACTGGTCAGCT	135
FASN	for CACGAACAACAGCCTCTT rev GCCTCCAGCACTCTACTA	171
LPL	for CACGAACAACAGCCTCTT rev ATCAGGAGAAAGGCGACTTGGAGC	159

**Table 2 animals-10-00818-t002:** miRNA-specific reverse transcription primers.

miRNA ID	Reverse Transcription Primers	miRNA-Special Primer
bta-miR-23b-3p	GTCGTATCCAGTGCAGGGTCCGAGGTATTCGCACTGGATACGACGTGGTAAT	TGCGGATCACATTGCCAGGGAT
bta-miR-126-3p	GTCGTATCCAGTGCAGGGTCCGAGGTATTCGCACTGGATACGACCGCATTAT	TGCGGCGTACCGTGAGTAAT
bta-miR-149-5p	GTCGTATCCAGTGCAGGGTCCGAGGTATTCGCACTGGATACGACGGGAGTGA	TGCGGTCTGGCTCCGTGTCTTC
bta-miR-24-3p	GTCGTATCCAGTGCAGGGTCCGAGGTATTCGCACTGGATACGACCTGTTCCT	TGCGGTGGCTCAGTTCAGCAG
bta-miR-199a-3p	GTCGTATCCAGTGCAGGGTCCGAGGTATTCGCACTGGATACGACTAACCAAT	TGCGGACAGTAGTCTGCACAT
bta-miR-193a-3p	GTCGTATCCAGTGCAGGGTCCGAGGTATTCGCACTGGATACGACACTGGGAC	TGCGGAACTGGCCTACAAAGT
bta-miR-199b	GTCGTATCCAGTGCAGGGTCCGAGGTATTCGCACTGGATACGACGAACAGAT	TGCGGCCCAGTGTTTAGACTAT
bta-miR-29b	GTCGTATCCAGTGCAGGGTCCGAGGTATTCGCACTGGATACGACAACACTGA	TGCGGTAGCACCATTTGAAATC
bta-miR-199a-5p	GTCGTATCCAGTGCAGGGTCCGAGGTATTCGCACTGGATACGACAACAGGTA	TGCGGCCCAGTGTTCAGACTA
bta-miR-379	GTCGTATCCAGTGCAGGGTCCGAGGTATTCGCACTGGATACGACCCTACGTT	TGCGGTGGTAGACTATGGAA
bta-miR-33a	GTCGTATCCAGTGCAGGGTCCGAGGTATTCGCACTGGATACGACTGCAATGC	TGCGGGTGCATTGTAGTTGC
bta-miR-449a	GTCGTATCCAGTGCAGGGTCCGAGGTATTCGCACTGGATACGACACCAGCTA	TGCGGTGGCAGTGTATTGTTA
bta-miR-148a	GTCGTATCCAGTGCAGGGTCCGAGGTATTCGCACTGGATACGACACAAAGTT	TGCGGTCAGTGCACTACAGAA
bta-miR-15a	GTCGTATCCAGTGCAGGGTCCGAGGTATTCGCACTGGATACGACACAAACCA	TGCGGTAGCAGCACATAATG
bta-miR-u6	AACGCTTCACGAATTTGCGT	CTCGCTTCGGCAGCACA

**Table 3 animals-10-00818-t003:** The statistical data of RNA sequencing.

Sample Name	Total Tag	Mapped Tag	Percentage (%)
b_AD_1	24,216,947	23,523,986	97.14
b_AD_2	24,608,755	23,928,858	97.24
b_AD_3	28,398,824	27,860,568	98.1
b_PAD_1	27,858,763	26,797,401	96.19
b_PAD_2	23,590,905	22,347,469	94.73
b_PAD_3	30,589,648	30,064,620	98.28

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
