# Peer review of "Isolation and Identification of Bovine Preadipocytes and Screening of MicroRNAs Associated with Adipogenesis"

_animals, 2020, doi:10.3390/ani10050818_

Round 1

Reviewer 1 Report

The authors addressed all my comments, hence recommended for publication

Author Response

Thanks for your suggestions, english language and style of the manuscript has been modified.

Reviewer 2 Report

Yu and colleagues have isolated and identified bovine preadipocytes, and screened miRNAs associated with adipogenesis. A total of 250 miRNAs were differentially expressed between preadipocytes and adipocytes. The paper is scientifically sound and the results are meaningful.

My major concern about the manuscript is that how many biological replicates are used for small RNA-seq?

.

Several minor comments that I would like to suggest to the authors are:

Line 16, I suggest to change “regulation” to “regulatory”.

Line 18, Change “the expression patterns…” to “the expression patterns of miRNAs…”.

Line 19-20, The statement that “Deep sequence analysis of miRNA libraries from preadipocytes and adipocytes allowed the identification of 63 novel and 67 known bovine miRNAs” is not correct, as the authors have identified 78, 71, and 48 novel miRNAs and 497, 491, and 524 known miRNAs in the preadipocytes, and 44, 54, and 47 novel miRNAs and 519, 522, and 504 known miRNAs in the adipocytes.

Line 54, in vitro in italic.

Line 76, I suggest to change “sponging” to “binding”.

Line 82, The literature 14 is not properly cited.

Line 101, When the reagents or instruments are mentioned for the first time, the company name, city, and country should all be marked. The authors should double-check the paper.

Line 33, line 145, line 150, what is the difference among “qPCR, qRT-PCR, and RT-qPCR”. Are all these abbreviations for quantitative reverse transcription PCR? The authors should use only one abbreviation in their paper.

Line 207, The version of SPSS software should also be noted.

Line 214, line 216, The figure 1 need to be relabeled. What do “Figure 1a-a,a-b” and “Figure 1a-c,a-d” mean?

Author Response

Responds to Reviewer:

Q1. My major concern about the manuscript is that how many biological replicates are used for small RNA-seq?

Response: Each of three biological replicates in preadipocyte group and mature adipocyte group were used for small RNA-seq.

Q2. Line 16, I suggest to change “regulation” to “regulatory”.

Response: Thanks for the suggestion.The spelling has been modified.

Q3. Line 18, Change “the expression patterns…” to “the expression patterns of miRNAs…”.

Response: The phrase has been modified.

Q4. Line 19-20, The statement that “Deep sequence analysis of miRNA libraries from preadipocytes and adipocytes allowed the identification of 63 novel and 67 known bovine miRNAs” is not correct, as the authors have identified 78, 71, and 48 novel miRNAs and 497, 491, and 524 known miRNAs in the preadipocytes, and 44, 54, and 47 novel miRNAs and 519, 522, and 504 known miRNAs in the adipocytes.  

Response:  Thanks for reminder. This part has been revised, as shown in line 19-21 of the revised manuscript.

Q5. Line 54, in vitro in italic.

Response: The format has been revised.

Q6. Line 76, I suggest to change “sponging” to “binding”

Response: The spelling has been modified.

Q7. Line 82, The literature 14 is not properly cited.

Response:  We have revised the literature 14.

Q8. Line 101, When the reagents or instruments are mentioned for the first time, the company name, city, and country should all be marked. The authors should double-check the paper.

Response: The information of reagents and instruments has been added in the paper.

Q9. Line 33, line 145, line 150, what is the difference among “qPCR, qRT-PCR, and RT-qPCR”. Are all these abbreviations for quantitative reverse transcription PCR? The authors should use only one abbreviation in their paper.

Response:  Thanks for suggestions. The three abbreviations are for quantitative reverse transcription PCR. The abbreviation has been revised in the paper.

Q10. Line 207, The version of SPSS software should also be noted.

Response: The version of SPSS software has been added in the paper.

Q11. Line 214, line 216, The figure 1 need to be relabeled. What do “Figure 1a-a, a-b” and “Figure 1a-c, a-d” mean?

Response:  Thanks for your reminder, The figure 1 has been revised.

Reviewer 3 Report

The manuscript “Isolation and Identification of Bovine Preadipocytes and Screening of MicroRNAs Associated with Adipogenesis” is very interesting and contributes to the understanding of bovine adipogenesis. I have several comments that should be addressed by the authors and I am especially concerned about method description.

In Simple Summary, it was stated that “These results can provide both technical support and a research basis for promoting bovine adipocyte fat deposition”, but this approach was not mentioned in the manuscript. How can bovine adipogenesis be artificially controlled based on the provided information?

Adipogenesis is one and not the “primary focus” on animal breeding, and muscle development is a research target too. The authors should consider that the primordial cells (mesenchymal stem cells) that differentiate in adipocytes are the same that can differentiate in muscle cells.

In vitro adipogenesis induction was used to study difference in miRNA expression levels. Does it reflect the condition observed in vivo?

Why cell cultures were performed in temperature inferior to normal bovine temperature (37 vs. 38.5-39 oC)?

How b-actin was validated as the reference gene? LPL and b-actin genes were mentioned in the text (lines 145 and 148), but Table 1 reports primers for DLK1 and GAPDH.

What primary antibody (line 154) was used? What did they detect?

Does the kit provides the control to perform the standard curve (lines 163-164)?

For RNA-seq, other small RNAs were depleted before sequencing miRNA?

Figure 1 presents two cell isolation methods (explant and digestion), but only the digestion method was presented in Methods.

In Figure 2, the images of immunofluorescence staining are very dark, making it difficult to observe the differences.

In Discussion, it was stated best induction condition with DMEM/F12 (line 373), but it’s necessary to highlight the conditions described in lines 233-234 of results (with 2% SR and 0.5% SFM).

Inform the sample (preadipocyte or adipocyte) where miR-129-5p was up-regulated (line 356). Do the same for all discussed miRNA.

In Discussion, pathways were described according to the role of the potential target genes on cascades. However, it is necessary to consider whether miRNAs were up- or down-regulated, leading to down- or up-regulation of target genes, and the consequences of this relationship to the pathways.

Author Response

Responds to Reviewer:

Q1. In Simple Summary, it was stated that “These results can provide both technical support and a research basis for promoting bovine adipocyte fat deposition”, but this approach was not mentioned in the manuscript. How can bovine adipogenesis be artificially controlled based on the provided information?

Response: We provided a standard protocol for isolating bovine preadipocytes from adipose tissue in the Materials and Methods. Then different induction conditions were used to get an optimal condition. Meanwhile, crucial miRNAs associated with adipocyte differentiation were screened. So, these results could provide technical support for future studies.

Q2. Adipogenesis is one and not the “primary focus” on animal breeding, and muscle development is a research target too. The authors should consider that the primordial cells (mesenchymal stem cells) that differentiate in adipocytes are the same that can differentiate in muscle cells.

Response: Thanks for the suggestions, we have revised in the paper. Because of different reagents used for myogenic differentiation and adipogenic differentiation, adipogenic differentiation of preadipocytes was induced under specific conditions in this study, induction of mesenchymal stem cells into muscle cells outside the scope of this study was not performed.

Q3. In vitro adipogenesis induction was used to study difference in miRNA expression levels. Does it reflect the condition observed in vivo?

Response: Adipogenesis induction in vitro can’t fully reflect the condition in vivo. To prove the influence of differently expressed miRNAs in bovine preadipocyte differentiation, we may perform subsequent animal experiments.

Q4. Why cell cultures were performed in temperature inferior to normal bovine temperature (37 vs. 38.5-39 ℃)?

Response: The classic culture condition is currently used by most researchers. Some other studies also used the temperature to culture bovine adipocytes, like Fernyhough ME, Vierck JL, Hausman GJ, Mir PS, Okine EK, Dodson MV. Primary adipocyte culture: adipocyte purification methods may lead to a new understanding of adipose tissue growth and development. Cytotechnology. 2004;46(2-3):163–172. doi:10.1007/s10616-005-2602-0.

Q5. How b-actin was validated as the reference gene? LPL and b-actin genes were mentioned in the text (lines 145 and 148), but Table 1 reports primers for DLK1 and GAPDH.

Response: We verified the expression levels of different reference genes in cattle tissues and found that the expression of β-actin was the most stable. And the β-actin was selected as the internal control because we read some manuscripts which have been published that also validated a gene associated with lipid metabolism, the β-actin was selected as internal control in paper, like that Jiang, et al. ,2018, The effect of short/branched chain acyl-coenzyme A dehydrogenase gene on triglyceride synthesis of bovine mammary epithelial cells. Arch. Anim. Breed. 61(1):115-122; Fang,et al., 2017 study, Identification of the bovine HSL gene expression profiles and its association with fatty acid composition and fat deposition traits.

Thanks for reminder. We have added the primers for DLK1 and LPL and modified the reference gene primers.

Q6. What primary antibody (line 154) was used? What did they detect?

Response: The antibody of DLK, PPARγ and LPL is used to detect the expression levels and subcellular localization by Immunofluorescence.

Q7. Does the kit provide the control to perform the standard curve (lines 163-164)?

Response: Yes. The standard curve is calculated by using the standard measurement in the kit. Briefly, the total protein concentration was measured with the Enhanced BCA Protein Quantitation Assay Kit following the manufacturer's protocols (KeyGEN BioTECH, KGP902, Jiangsu, China). The experiment was repeated 3 times, and blank group was established as a reference. the blank group was added ddH2O replace the protein sample. We need to draw of the standard curve before measure the concentration of each protein sample following the protocol. Each tube is added according to the manufacturer’s protocol.

Q8. For RNA-seq, other small RNAs were depleted before sequencing miRNA?

Response: miRNA sequencing of preadipocytes before and after differentiation, only 18-30nt RNAs segments, which included miRNA, pi-RNA, tsRNA and siRNA, were separated by PAGE gel to filter small RNA.

Q9. Figure 1 presents two cell isolation methods (explant and digestion), but only the digestion method was presented in Methods.

Response: Thanks for your reminder, we have added the explant method in Methods.

Q10. In Figure 2, the images of immunofluorescence staining are very dark, making it difficult to observe the differences.

Response: We have modified the picture to make it clearer.

Q11. In Discussion, it was stated best induction condition with DMEM/F12 (line 373), but it’s necessary to highlight the conditions described in lines 233-234 of results (with 2% SR and 0.5% SFM).

Response: Thanks for the suggestion, we have revised Discussion.

Q12. Inform the sample (preadipocyte or adipocyte) where miR-129-5p was up-regulated (line 356). Do the same for all discussed miRNA.

Response: Thanks for the suggestion. We have modified in the manuscipt.

Q13. In Discussion, pathways were described according to the role of the potential target genes on cascades. However, it is necessary to consider whether miRNAs were up- or down-regulated, leading to down- or up-regulation of target genes, and the consequences of this relationship to the pathways.

Response: In discussion, the target relationships between miRNAs and genes were hypotheses. Subsequent experiments, like target relationship verification (DLK1 and miR-129-5p), function of target gene and pathway (ANGPTL8, NR1H3, STAR) are on-going.

Round 2

Reviewer 3 Report

The authors addressed all the suggestions and comments, and the revisedmanuscript is suitable for publication.

This manuscript is a resubmission of an earlier submission. The following is a list of the peer review reports and author responses from that submission.

Round 1

Reviewer 1 Report

In this study, the authors explored the molecular regulatory mechanism of adipocyte differentiation and adipocyte formation, bovine preadipocytes were isolated and induced into adipocytes, then the expression patterns between preadipocytes and adipocytes were detected through RNA sequencing. Deep sequence analysis of miRNA libraries from preadipocytes and adipocytes identified 63 novel and 67 known bovine miRNAs. The study is well planned and nicely presented, however my major concerns are as following

Primers made in house? Where they validated previously for specificity only to the mature miRNA and not the pri- or pre-?  Typically miRNA primers have hair pin loops, can authors confirm validity and function of their primers?

How were bovine cells validated as pre-adipocytes?

Line 319-320 (Figure 5) how you find the expression trend of the target genes? qPCR, RNA-seq or just prediction?

Discussion 321-336-this para looks like introduction. Line 337-349-what is the relevance of this para with the current study. Why the author emphasizing on the different methods of adipocyte differentiation? The authors must focus on the adipogenic markers (differentially expressed mRNAs and miRNAs) in different stages of proliferation and differentiation. The discussion needs revision.

Many figures looked smashed and hard to read. Should be reformatted.

Reviewer 2 Report

The manuscript reported microRNAs related to differentiation of bovine preadipocytes by RNAseq. The findings are able to provide a new insight to understand molecular mechanisms underlying differentiation of bovine preadipocytes.

Despite the potential of the work, several issues will need further improvement.

In simple summary and abstract, several sentences have vague expression such as plenty of genes, cellular and metabolic process, and many novel miRNAs. They should be revised for improving the clarity of the sentences. Introduction section fails to deliver the sufficient background information, study question, and hypothesis. Please improve the introduction section. Please explain the rationale to establish a standard protocol for isolating and identifying bovine preadipocytes. What are the differences between the protocol used in this study and the current method for isolating and identifying bovine preadipocytes. Result section from 3.1 to 3.3 does not seem to be necessary. It is believed that only one figure to confirm successful adipogenesis is enough.

Reviewer 3 Report

1.Overall, the study is well thought out and conducted properly. However, there are many places in the document where procedures and justification is not specific enough or slightly inaccurate.

2. While there is very limited bovine relation to the study, a glaring error in the introduction states that subcutaneous fat(back fat) is an indicator of animal quality. This is not true, but more an indicator of how well that animal was managed during the finishing phase, and that animals genetic predisposition to muscling or fattness.

3. While i do know this was a microRNA paper looking at adipogenesis, the paper severly lacks any real implications or potential future applicablity. Who will this potentially benefit and how?
